# MODELING FAKE NEWS IN SOCIAL NETWORKS WITH DEEP MULTI-AGENT REINFORCEMENT LEARNING

## ABSTRACT

Over the last few years deep multi-agent reinforcement learning (DMARL) has become an increasingly active area of research with hundreds of papers submitted to top machine learning conferences every year. However, so far there have been few real world use cases that benefited from the progress in the field. We use DMARL to develop a flexible computational model of fake news on social networks in which agents act according to learned best response functions. We achieve this by extending an information aggregation game to allow for fake news and by representing agents as recurrent deep Q-networks (DQN). In the game, agents repeatedly guess whether a claim is true or false taking into account an informative private signal and observations of actions of their neighbors on the social network in the previous period. We incorporate fake news into the model by adding an adversarial agent, the attacker, that either provides biased private signals to, or takes over, a subset of agents. The attacker can follow either a hand-tuned or trained policy. Our model allows us to tackle questions that are analytically intractable in fully rational models, while ensuring that agents follow reasonable best response functions. Our results highlight the importance of awareness, privacy and social connectivity in curbing the adverse effects of fake news and open an entire new real world application area for DMARL.

## 1 INTRODUCTION

**Problem setting**   At least since the 2016 US presidential election, fake news on social networks, aimed at manipulating the users' perception of facts, has been recognized as a major issue in open societies. Yet, to date little is understood about how fake news spread on social networks, when a piece of fake news is effective in swaying public opinion and what interventions might be successful in mitigating the effect of fake news. Our lack of understanding of this important phenomenon is in large part due to the difficulty of modeling complex decision making processes on social networks. To tackle this issue, we extend a standard game of information aggregation on a social network, see Mossel et al. (2015), to accommodate fake news and solve this, analytically intractable, game using deep multi-agent reinforcement learning. This approach is flexible, computationally tractable, and allows us to work towards answering the questions raised above.

**Related work**   Our work relates to two main strands of literature: (i) models of information aggregation in social networks and (ii) deep multi-agent reinforcement learning as a tool to solve games. The standard information aggregation game, outlined in Section 2.1, involves agents (repeatedly) guessing an unknown state of the world taking into account an informative private signal and observations of actions of their neighbors on the social network in the previous period. Note that this model involves *no fake news*. There are two main approaches to studying this game. In the rational agent approach, agents' decision are best responses and the analysis focuses on the Nash equilibria of the game. In an important contribution, Mossel et al. (2015) show that under certain conditions on the social network, in the limit $T \to \infty$ (number of periods) and $n \to \infty$ (number of agents), agents will converge and will agree on the correct state of the world. This approach is not tractable for finite $T$ and $n$ and therefore does not allow, for example, the study of transient behavior. An alternative approach to studying the game of information aggregation is heuristic, see Golub & Sadler (2016) for a review of many important contributions. While heuristic approaches are more flexible to changes in the model setup and allow the study of transient behavior, agent strategies can be too

simplistic. For example, agents may treat repeated observations of a neighbor's guess as conveying additional information, even when this is not the case (this occurs for example when this neighbor's only information is his private signal). Furthermore, heuristic behavior does not adapt to changing circumstances and would therefore not allow us to study how agent behavior changes in the presence of fake news. Our deep multi-agent reinforcement learning approach combines both worlds. Our approach is flexible since it does not require the model to be analytically tractable and finds near-optimal policies through reinforcement learning.

Our work is also related to studies of competitive information diffusion, see Alon et al. (2010); Etesami & Başar (2014). However, while the literature on competitive information diffusion focuses on the strategic seeding of a pre-determined diffusion process, our work focuses on modeling the information diffusion process itself through the agents' learned actions.

Deep multi-agent reinforcement learning (DMARL) has been applied to solve games of different structure. Policies or value functions are approximated by deep neural nets and trained via backpropagation. Some recent examples are Letcher et al. (2018); Foerster et al. (2018); Lerer & Peysakhovich (2017); Balduzzi et al. (2018). Often the focus is on methods that allow agents to converge on a particular, socially desirable policy, such as cooperation in social dilemmas without changing the payout structure. Since this is not an issue for most of our analyses, we make use of Independent Q-learning, see Tampuu et al. (2017). However, for some of our more advanced analyses, convergence does become an issue and applying more advanced DMARL methods provides a promising avenue of future research.

**Contribution** To the best of our knowledge, we are the first to develop a flexible computational model of fake news on social networks in which agents act according to learned best response functions. We achieve this by extending a standard information aggregation model to allow for fake news and by representing agents as recurrent deep Q-networks (DQN). We incorporate fake news into the model by adding an adversarial agent, the attacker, that either provides biased private signals to or takes over a subset of agents. We conduct a number of experiments to answer three main questions. (i) What determines the effectiveness of the attacker in reducing the accuracy of information aggregation? (ii) How well can agents learn adapt to the presence of an attacker? (iii) When attacker and agents learn simultaneously, how do policies evolve over time? Our results suggest three potential interventions to curb the effectiveness of fake news on social networks. First, "vaccinate" users of social networks by making them aware of the presence of fake news. Second, keep private information on social networks private so that attackers cannot target poorly informed or well connected users. Third, encourage the formation of "balanced" rather than clustered or fragmented social networks, so that information can "flow" effectively across the network.

Lastly, we contribute to the MARL literature by providing an important and concrete real-world application of methods that to date have found limited practical application.

## 2 BACKGROUND

### 2.1 AN INFORMATION AGGREGATION GAME ON A SOCIAL NETWORK

We consider a sequential game of information aggregation over $T$ periods indexed by $t = 1, \ldots, T$ with a set of agents $N$, see Mossel et al. (2015); Golub & Sadler (2016) for details. Agents interact via a fixed, possibly directed, graph $\mathcal{G}(N, E)$, where $E$ denotes the set of edges connecting the nodes $N$; see Figure 1 for an illustration of the networks used in this paper. For concreteness, one can think of the agents as users of a social network, such as Twitter. A directed link from agent A to agent B would then imply that A follows B on Twitter.

Agents are presented with a claim that is either true or false. Again, for concreteness, one can think of the claim as a factual statement encountered in a piece of news that has been shared among the agents on the social network.[1] True claims are denoted as $\theta = 1$ and false claims as $\theta = 0$. True and false claims occur with equal probability; that is, the unconditional probability of a claim being

---

[1] We restrict our analysis to the case when claims are factual statements that can be objectively verified as true or false. Of course this prevents us from studying the dynamics of opinions. However, since we focus on the effect of fake news on the social network, restricting our analysis to factual statements is natural.

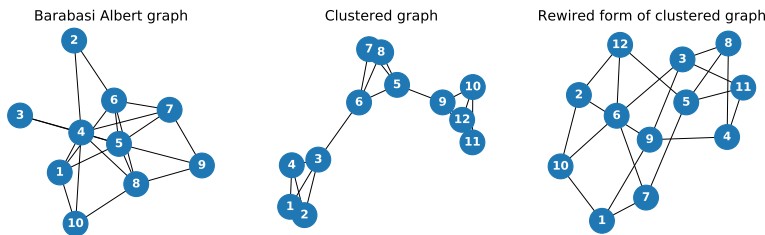

Figure 1: Left: Instance of a Barabsi-Albert random network with $|N| = 10$ that was used in our analyses (unless otherwise stated). Middle: stereotypical clustered network with $|N| = 12$ with three ($n_c = 3$) fully connected clusters of four nodes ($s_c = 4$) each. Right: the clustered network in balanced form produced by random rewiring of link pairs (preserves degree distribution). These networks were used in our analysis of the effectiveness of information aggregation and attack (i) in clustered vs. balanced networks and (ii) for spread out vs. focused attack strategies.

true or false is $\mathbb{P}(\theta = 0) = \mathbb{P}(\theta = 1) = 1/2$. This choice simplifies our analysis without affecting any of our qualitative results. Agents receive private signals $s^i \sim F_\theta$ once at the beginning of an episode. In the baseline $F_\theta$ is such that private signals are always informative but noisy. In each period agents choose a binary action $a_t^i \in \{0, 1\}$. Agents observe the actions of their neighbors on the graph from the previous period. At the end of the episode, agents are rewarded for actions that matched the veracity of the claim presented. For a given claim $\theta$ and sequence of actions $\{a_t^i\}$, an agent's total discounted reward is $r_i = \sum_t^T \gamma^t \mathbb{1}\{a_t^i = \theta\}$.

Note that in this game, for $t < T$, it is not necessarily optimal to choose the action that matches the maximum of the posterior likelihood of $\theta$. This is because an agent wants to prevent others from simply copying its action, since that reveals no information about the others' private signal. Injecting noise into actions may therefore be beneficial. Despite these concerns, it can be shown that, under certain conditions on the network, in the limit $T \to \infty$ (number of periods) and $n \to \infty$ (number of agents), agents will converge and will agree on the correct $\theta$, see Mossel et al. (2015).

## 2.2 MULTI-AGENT REINFORCEMENT LEARNING: INDEPENDENT Q-LEARNING

In principle, reinforcement learning (RL) can be used to learn the agents' policy (or response functions) in any game. Many methods have been proposed to learn reasonable best response functions in various multi-agent reinforcement learning (MARL) settings, see Foerster et al. (2016); Letcher et al. (2018); Lerer & Peysakhovich (2017). In this paper, the agents' response functions will be learned via independent Q-learning (IQL) and Q functions will be approximated by deep recurrent neural networks, see Tampuu et al. (2017); Foerster et al. (2016). We relegate a discussion of the RL and IQL framework to Appendix A.1.

## 3 METHODS

**Graphs** We consider three types of *undirected* graphs, see Figure 1 for an illustration. An instance of a Barabasi-Albert random graph with $|N| = 10$, see Barabási & Albert (1999); a clustered graph with three clusters ($n_c = 3$) of fully connected nodes with four nodes each ($s_c = 4$) such that $|N| = 12$; and a balanced graph ($|N| = 12$) that is obtained by randomly rewiring the clustered graph. Random rewiring involves repeatedly selecting two edges uniformly at random and swapping the terminal nodes of these edges. This preserves the degree sequence, i.e. the number of neighbors of each node, but removes the clusters such that the rewired graph is "balanced". The Barabasi-Albert random graph is often used to model social networks and is therefore a natural choice for our application. The clustered graph is a stereotypical example of a social network which is comprised of cliques that are weakly connected. It thereby represents a more "fragmented" or "polarized" social network. The balanced graph removes these clusters from the graph while maintaining the degree sequence and can thus help isolate the effect of clusters on information aggregation.

**Fake news** We extend the model of information aggregation in order to study the effects of fake news on information aggregation. We introduce an adversary, the so called attacker, who wants to reduce the other agents' total reward $\sum_i r_i$. To disambiguate between the attacker and other agents, we often refer to these other agents as citizens. The attacker wants to persuade citizens to support false claims and dissuade them from supporting true claims. The attacker can achieve this in one of two ways. First, in the *biased signal attack*, he can distribute a fixed budget of bias $\beta = \sum_i \beta^i$ across a subset of the agents such that citizen $i$'s private signal is drawn from $s^i \sim F_\theta(\beta_i)$.[2] We choose $s^i \sim \mathcal{N}(\theta + \beta^i(1 - 2\theta), \sigma^2)$. We set $\sigma^2 = 1$ throughout. In this case the choice of $\{\beta_i\}_{i \in N}$ is *hand-tuned*, i.e. set by the experimenter and not optimized. Second, in the *agent takeover attack*, he can directly take over an agent and act on his behalf. In this case the attacker optimizes his attack policy by IQL.

**Learning to aggregate information** We hold the network fixed across episodes. At the beginning of each episode $\theta$ and all private signals are drawn at random according to the distributions given above. For *citizens* we train a single DQN. In period $t$, the citizen observes the episode private signal $s^i$, his and the actions of his neighbors in the previous period $\{a_{t-1}^j\}_{j \in B_i}$ ($B_i$ is the set of neighbors of agent $i$ augmented by $i$) and the agent id $i$ which identifies the agent's network position. To maintain a constant length input vector, unobserved actions are encoded as $-1$. Thus, the corresponding DQN is a function $Q : \mathbb{R} \times \{-1, 0, 1\}^{|N|} \times N \times \{0, 1\} \mapsto \mathbb{R}$.

In the *agent takeover attack*, the attacker is represented by a DQN. This DQN is distinct from the citizen DQN. At the beginning of each episode the attacker chooses a citizen uniformly at random from $N$ and acts on his behalf. In period $t$, the attacker observes $\theta$ without noise, his and the actions of his neighbors in the previous period $\{a_{t-1}^j\}_{j \in B_i}$ and the agent id $i$. Thus, the corresponding DQN is a function $Q : \{0, 1\} \times \{-1, 0, 1\}^{|N|} \times N \times \{0, 1\} \mapsto \mathbb{R}$. In the *biased signal attack*, the attacker is *not* represented by a DQN. Instead, the biases are hand-tuned. At the beginning of each episode the attacker chooses a set of citizens uniformly at random from $N$ and delivers biased signals to them as outlined above.

**Measuring information aggregation** Intuitively, agents should be able learn to extract information from their private signals and their neighbors' actions, such that at the end of an episode their expected reward ($\mathbb{E}[r_T^i]$) should be significantly higher than at the beginning of an episode ($\mathbb{E}[r_0^i]$). That is, agents should aggregate information over time. In the following, we give a formal definition of information aggregation for finite $N$ and $T$ that is useful in our case.

**Definition 1** (Information aggregation). Let $\mathcal{F} = \{s^i\}_{i \in N}$ denote the set of all realized signals in a given episode. Let $\hat{\theta} = \text{argmax}_\theta \mathbb{P}[\theta \mid \mathcal{F}]$ denote the maximum-a-posteriori estimator of $\theta$ given $\mathcal{F}$. Then information is aggregated if for all $i \in N$, we have $a_T^i = \hat{\theta}$.

That is, under this definition information is aggregated if agents act as if they had seen all private signals. In reality, it is not reasonable to expect perfect information aggregation with actions learned via independent Q-learning. We therefore define the following measure of information aggregation.

**Definition 2** (Accuracy of information aggregation). The accuracy of information aggregation at time step $t$ is $A_t = \mathbb{E}[\mathbb{1}\{a_t^i = \theta\}]$.

A useful benchmark to compare $A_t$ to is of course $\mathbb{E}[\mathbb{1}\{\hat{\theta} = \theta\}]$. Another useful benchmark is $\mathbb{E}[\mathbb{1}\{\tilde{\theta} = \theta\}]$, where $\tilde{\theta} = \text{argmax}_\theta \mathbb{P}[\theta \mid s^i]$ is the MAP estimator given only a single private signal. $A_T$ can never exceed $\mathbb{E}[\mathbb{1}\{\hat{\theta} = \theta\}]$ and $A_1$ can never exceed $\mathbb{E}[\mathbb{1}\{\tilde{\theta} = \theta\}]$ and of course $\mathbb{E}[\mathbb{1}\{\hat{\theta} = \theta\}] > \mathbb{E}[\mathbb{1}\{\tilde{\theta} = \theta\}]$ provided $|N| > 1$. Once we have trained the agents via Q-learning it is easy to estimate $A_t$ by evaluating a set of batches of games $B$ and computing $\hat{A}_t = 1/|B|1/|N| \sum_{b \in B, i \in N} \mathbb{1}\{a_{t,b}^i = \theta\}$, where $a_{t,b}^i$ is the action taken by agent $i$ in batch $b \in B$. In the following we will simply refer to $\hat{A}_t$ as the accuracy.

---

[2] Biased signals can be thought of as targeted ads to users of the social network. Indeed, it been shown that fake news attacks have taken exactly this form, see Chiou & Tucker (2018). A fixed bias budget corresponds roughly to an attacker with a fixed budget for targeted social media ads.

**Heuristic benchmark**    We compare the accuracy of the DQN against the DeGroot heuristic, De-Groot (1974). Under this heuristic information aggregation rule at $t = 0$, we set $a_0^i = \tilde{\theta}(s^i)$. In each subsequent period actions are set to $a_t = U a_{t-1}$, where $a_t$ is the vector of actions taken by citizens and $U = D^{-1}(I + M)$, where $I$ is the identity matrix, $M$ is the graph adjacency matrix and $D$ is a diagonal matrix with entries $D_{ii}$ being the number of neighbors of node $i$ plus one (i.e. a self loop). Information is therefore aggregated through repeated averaging of past actions.

## 4    EXPERIMENTS

Our objective is to improve our understanding of the propagation of fake news in social networks and discover potential interventions to curb its effectiveness. This goal translates into three main questions. (i) What determines the effectiveness of the attacker in reducing the accuracy of information aggregation? In particular, how does the network structure and the choice of the attacked agent affect attack effectiveness? (ii) How well can agents learn adapt to the presence of an attacker? (iii) When attacker and citizen learn simultaneously (agent takeover attack), how do policies change over time? To study these questions we set up a series of training and testing scenarios.

**Training scenarios:**    We implement the following training scenarios. (i) **Baseline:** We train citizens in the absence of any attack, i.e. all citizens receive unbiased signals and no citizen is taken over by the attacker. We train baseline models for the Barabasi-Albert, clustered and balanced graphs. This scenario establishes a benchmark for the ability of IQL agents to learn to aggregate information in the absence of attack. (ii) **Biased signal attack:** We train citizens in the presence of a biased signal attack where a single, randomly chosen citizen receives a biased signal ($\beta = 3$). We train this model for the Barabasi-Albert graph only. This scenario allows us to evaluate whether citizens can learn to adapt to the presence of a biased signal attack and thereby mitigate the adverse effects of the attack. (iii) **Agent takeover attack:** We train citizens and attacker simultaneously. We train this model for the Barabasi-Albert graph only. This scenario allows us to understand the effectiveness of attack as it co-evolves with the citizen's effort to mitigate the attacker's ability to manipulate citizens. Note that IQL may not converge to stable policies in such a game. It is interesting however, to understand the action dynamics that emerge under IQL as they can help us understand the co-evolution of fake news attack and defense in real social networks. (iv) **Random action attack:** We train citizens in the presence of an attacker that selects a single citizen uniformly at random and then, in each period, picks an action uniformly at random. We train this model for the Barabasi-Albert graph only. This scenario serves as a benchmark for the agent takeover attack. The accuracy under the random action attack should exceed the accuracy under the agent takeover attack.

**Testing scenarios:**    Given the models obtained for these different training scenarios we conduct a number of testing scenarios. First, for each attack training scenario, we evaluate accuracy, as defined above, both in the presence of the corresponding attack (as trained) and in the absence of any attack. For the baseline scenario we consider two testing scenarios in addition to testing in the absence of any attack (as trained). First, we evaluate accuracy in the baseline model for the Barabasi-Albert graph in the presence of a biased signal attack with a single attacked agent and $\beta = 3$. Second, we evaluate accuracy in the baseline model for the clustered and balanced graphs under two biased signal attack scenarios. In the *focused* scenario, a single, randomly chosen citizen receives a biased signal with $\beta^i = 3$. In the *spread* scenario, two, randomly chosen citizens receive biased signals with $\beta^i = 1.5$ each.

## 5    RESULTS

Our experiments yield four main results, which we will discuss in turn.

**Baseline information aggregation:**    In the absence of attack, citizens learn to aggregate information with an accuracy close to the optimal benchmarks $\mathbb{E}[\mathbb{1}\{\hat{\theta} = \theta\}]$ as $t \to T$ and $\mathbb{E}[\mathbb{1}\{\tilde{\theta} = \theta\}]$ for $t = 0$. Our method's accuracy also substantially outperforms the accuracy achieved by DeGroot information aggregation heuristic. This highlights the appropriateness of IQL to solve the standard information aggregation game. In the presence of an attack, information aggregation is severely disrupted. These results are illustrated in Figure 2 (A).

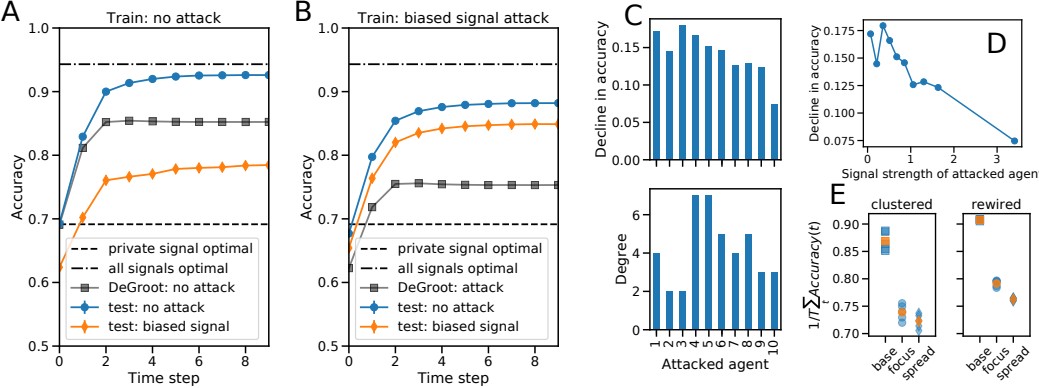

Figure 2: (A-B): Information aggregation over time in Barabasi-Albert graph. (A) Baseline scenario. Under the attack test scenario, a single agent receives a biased signal. (B) Biased signal attack scenario. The upper and lower dashed lines corresponds to the benchmarks $\mathbb{E}[\mathbb{1}\{\hat{\theta} = \theta\}]$ and $\mathbb{E}[\mathbb{1}\{\tilde{\theta} = \theta\}]$ respectively. Square markers correspond to the accuracy achieved by the De Groot information aggration heuristic in the baseline and biased signal attack scenarios respectively. (C-D) Agents are trained in the absence of any attacker on a Barabasi-Albert graph. (C) Top: Average decline in accuracy conditional on agent id which determines network position. Bottom: degree of attacked agent. (D) Effectiveness of conditioning on attacked agent signal strength. (E) Agents trained in the absence of any attacker for clustered (left) and rewired / balanced (right). Base: no attack. Focus: a single, randomly chosen agent receives a strong biased signal ($\beta^i = 3$). Spread: two, randomly chosen agents receive weak biased signals ($\beta^i = 1.5$). Blue markers represent training runs with different seeds for a given network and attack scenario. Orange markers are averages over the different training runs. Error bars computed over outcomes with different random seeds but fixed neural network weights are too small to be visible on the graphs.

**Determinants of effectiveness of attack:** Our experiments allow us to study the following determinants of attack effectiveness: network position of the attacked agent, signal strength of the attacked agent, network structure (clustered vs balanced) and distribution of bias across agents (focus vs spread). Let us consider each in turn.

To study the effect of network position, we compute the accuracy in the final time step ($t = T$) conditional on the network position of the attacked agent when evaluating the baseline model under the biased signal attack in the Barabasi-Albert network. Denote this conditional accuracy by $\hat{A}_T(i)$, where $i$ is the attacked agent. Let $\hat{A}_T$ denote the baseline accuracy in the absence of attack. We then define the decline in accuracy relative to baseline as $\Delta A(i) = \hat{A}_T - \hat{A}_T(i)$. We plot $\Delta A(i)$ as a function of $i$ in Figure 2 (C). It is clear that there is substantial heterogeneity in the attack effectiveness. As the tight correlation between $\Delta A(i)$ and the degree of $i$ in Figure 2 (C) shows, the heterogeneity in $\Delta A(i)$ arises from the network structure which implicitly gives more influence to those nodes with more neighbors (higher degree). If an attacker is able to target the biased signal to such highly connected nodes, his attack will be more effective.

Next we study the effect of the signal strength of attacked agent. Again, we restrict ourselves to evaluating the baseline model under the biased signal attack in the Barabasi-Albert network. A natural measure of a signal's strength, or informativeness, is the absolute value of the log likelihood ratio of the two states of $\theta$ conditional on the signal. Let $f(\theta \mid s)$ denote the posterior distribution over $\theta$ given the signal $s$. Then, the signal strength is given by $L(s) = |\log[f(\theta = 1 \mid s)/f(\theta = 0 \mid s)]|$. Let $L_k$ denote the $k$th decile of the empirical distribution of signal strengths for a particular experiment. We define the conditional accuracy $\hat{A}(L_k)$ as the average accuracy in the final time step conditional on the attacked agent's signal strength lying in $[L_k, L_{k+1}]$. $\Delta A(L_k)$ is defined analogously to $\Delta A(i)$. We plot $\Delta A(L_k)$ against $L_k$ in Figure 2 (D). There exists a strong negative correlation between attack effectiveness as measured by $\Delta A(L_k)$ and the attacked agent's signal strength. This is intuitive. If an agent has received a strong private signal in support of some value of $\theta$, a larger bias will be required to convince him of the contrary. Thus, if an attacker can target the biased signal to agents with weak private signals, his attack will be more effective.

Table 1: Test accuracy of agents in the final time step ($t = T = 10$) for different training and testing scenarios. All scenarios were run with the Barabasi-Albert graph. At test time without attack all agents receive unbiased signals and no agent is attacked. At test time with attack agents are exposed to the attack scenario they were trained under (except in the baseline case where attack is biased signal with $\beta = 3$).

| Training scenario | Testing accuracy ($t = T$) | |
|---|---|---|
| | Without attack | With Attack |
| Baseline (train: no attack, test: hand-tuned $\beta = 3$) | 0.926 | 0.784 |
| Biased signal attack ($\beta = 3$) | 0.882 | 0.849 |
| Random action attack | 0.909 | 0.908 |
| Agent takeover attack | 0.863 | 0.844 |

Lastly, we consider the case of clustered vs rewired/balanced networks and spread our vs focused attacks. This analysis is done evaluating the baseline model under the biased signal attack in the clustered and balanced networks. A number of results are worth noting. As can be seen in Figure 2 (E), for each scenario (baseline/no attack, focus and spread attacks) accuracy in the balanced network exceeds accuracy in the clustered network. This suggests that information aggregation is more effective in the balanced network. This is intuitive as the balanced networks has a shorter maximum path length between any two nodes thereby allowing information to "propagate" faster between nodes. In Figure 2 (E), we can also see that variation in the accuracy between training runs with different seeds (each run corresponds to one marker for a particular scenario) is smaller for the balanced network. This suggests that in the balanced network learning good policies is an easier task than in the clustered network. We can also see from Figure 2 (E) that spread attacks are more effective both in the clustered and balanced networks.

To summarize, targeting highly connected agents in the network or those with weak private signals, makes biased signal attacks more effective. Clustered networks are more susceptible to attack than balanced networks and spread attacks are more effective than focused ones.

**Adapting to attacks:** So far we have evaluated models trained in the absence of attack. However, one can expect that users of social networks will adapt over time to the presence of fake news. We therefore investigate to what extent citizens can learn to mitigate the effect of an attacker. For this purpose we train citizens in the presence of a biased signal attacker, a random attacker and an agent takeover attacker for the Barabasi-Albert network. We then evaluate the accuracy of each of these models in the absence of an attack and in the presence of the attack under which they were trained. We summarize our results in Table 1.

Let us first contrast the accuracy of the baseline model (trained without attack) under a biased signal attacker with the accuracy of a model trained under this attack scenario. When trained in the presence of an attacker the test accuracy under attack increases from $0.78$ to $0.84$ relative to the baseline. This can also be seen in Figure 2 (B). We conclude that agents can indeed learn to adapt to the presence of fake news. However, this adaptation comes at a cost. When trained in the presence of an attacker the test accuracy without attack decreases from $0.92$ to $0.88$ relative to the baseline. We conjecture that this is because agents learn to trust strong private signals less and are less likely to follow their neighbors' actions. A similar picture can be seen for the random attacker and the agent takeover attacker. To summarize, agents can learn to adapt to the presence of an attacker which reduces attack effectiveness. However, adaptation comes at a cost. If trained under attack and tested in the absence of attack, accuracy is lower than in the baseline which is trained without attack.

**Dynamics of attack and defense:** We now take a closer look at the agent takeover attack. Recall that in this scenario, a separate DQN agent is trained to minimize the total reward of the citizens by acting on behalf of a randomly chosen agent. This attacker knows $\theta$ and must then choose an action to convince the citizens of the opposite. Simply choosing the opposite action would be a naive strategy that is likely to be detected quickly by the citizens as they learn in the presence of this strategy. Always choosing a random action in every period, as the random attacker does, is also

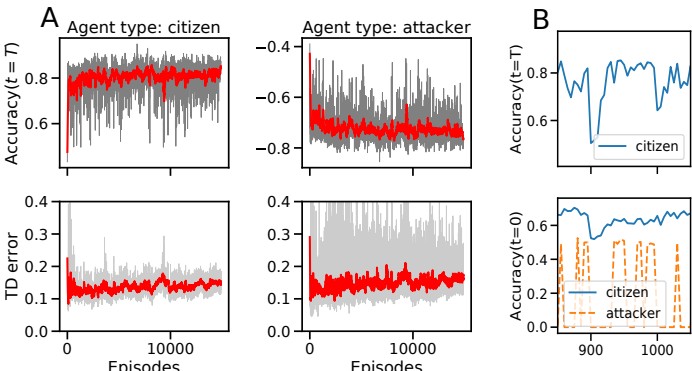

Figure 3: Dynamics of co-evolving attacker and citizen agents on a Barabsi-Albert graph. (A) Red lines are exponentially weighted moving averages. Top: accuracy in the last time step ($t = T = 10$) over episodes. Bottom: TD error of Q learning. (B) Close up of accuracy and actions of citizens and attacker early on during training averaged over multiple batches. Top: citizen accuracy in the last time step ($t = T = 10$). Bottom: citizen and attacker accuracy in the first time step ($t = T = 0$).

quite easily detectable by the citizens. This intuition is confirmed by results reported in Table 1 and Figure 3.

From Table 1 we see that the trained attacker clearly "outperforms" (i.e. more effective attack) the random action attack benchmark. The bottom panel of Figure 3 (B) shows the accuracy of the attacker in the first time step as a simple summary statistic of its strategy. The accuracy is either $0.5$ indicating that the attacker "randomizes" or $0$ indicating that the attacker "lies". While Q-learning only allows for deterministic policies the attacker can learn to lie if the attacked agent is in one partition of $N$ and be truthful if the attacked agent is in another partition. Since the attacked agent is chosen at random, this strategy amounts to a randomized attack strategy.

Note further, that neither the attacker's nor the citizens' policy converges as can been seen in the highly variable reward and unstable TD-error in Figure 3 (A) . Indeed, as can be seen in the bottom panel of Figure 3 (B) , the attacker's policy fluctuates between randomization and lying between episodes. These fluctuations can be seen as irregular "cycles" in the policy space of the attacker and the citizen. One such cycle in Figure 3 (B) shows the attacker switching to a lying strategy, while the citizens lose "trust" in their private signal (they nearly ignore it in the first time step). The result of these two events coming together can lead to substantial, temporary declines in accuracy.

We don't claim that these dynamics are accurate representations of the learning dynamics between users of social networks and pushers of fake news. However, we consider the failure of IQL to converge and the irregular dynamics that emerge more of a feature than a bug. Indeed these dynamics suggest two interesting, qualitative take aways. First, attackers and citizens are unlikely to converge to stable policies in real life. We should expect a continuing cycle of adaptation with periods of a high prevalence of fake news followed by periods of low prevalence. Second, the co-evolution of attackers and citizens can have unexpected effects on the citizens' ability to aggregate information in the social network. As we observe in Figure 3 (B) , the presence of neighbors that do not act in accordance with their private signals (i.e. the attacked agents) can lead citizens to learn to mistrust their own private signals.

## 6 CONCLUSION AND DISCUSSION

The deliberate manipulation of the public's perception of facts via fake news, in particular on social networks, has become a growing concern for policy makers and technology companies alike. Despite its importance, little is understood about the spread of fake news on social networks. This is in large part due to the technical difficulties involved in studying complex decision making behavior on social networks. We seek to close this gap by developing the (to the best of our knowledge) first framework for the study fake news on social networks that is theoretically grounded yet com-

putationally tractable and flexible. We achieve this by (i) extending a standard game of information aggregation to accommodate fake news and (ii) applying state-of-the-art deep multi-agent reinforcement learning to solve the game.

Our findings suggest a number of interventions that could contribute to making fake news attacks less effective. First, if agents are made aware of the presence of fake news, they can learn to adapt and mitigate its effect. However, this adaptation is likely to be accompanied by evolving attack strategies, so any adaptation will not be lead to permanent mitigation. In addition, the adaptation is likely to harm information aggregation in the absence of fake news. Second, keeping private information on social networks private, can make it harder for attackers to condition their attacks on network position and general informedness (as the real world analogue of signal strength). This should reduce attack effectiveness. Third, encouraging well balanced social networks can improve information aggregation in general and make fake news attacks less effective in particular.

Our current approach has two main shortcomings. First, it does not scale well to larger populations of agents since it requires the entire social network to be trained simultaneously. One way forward is to train agents in sub-graphs feeding them an embedding of the graph, see for example Gilmer et al. (2017), and then composing larger graphs at test time. The second shortcoming is the use of IQL in the agent takeover attack scenario. While the lack of convergence leads to interesting qualitative insights, MARL methods that encourage convergence of policies would be desirable here, at least as benchmarks. One possible way to achieve this would be to apply stable opponent shaping to citizen and attacker, see Letcher et al. (2018).

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

## A  APPENDIX

### A.1  MULTI-AGENT REINFORCEMENT LEARNING

**Single-Agent Reinforcement Learning**  Consider an MDP specified by the following tuple: $\langle S, A, P, r, \gamma \rangle$. In the fully observable case the environment has an observed true state, $s \in S$, and at each time step the agent chooses an action $a \in A$, which induces a probabilistic transition to the next state via the state transition function $P(s'|s, a) : S \times A \times S \rightarrow [0, 1]$. The reward function assigns an instantaneous reward to each state-action pair, $r(s, a) : S \times A \rightarrow \mathbb{R}$, and $\gamma \in [0, 1)$ is a discount factor.

In single agent RL the task of the agent is to maximize the discounted reward per episode, $R_t = \sum_{l=0}^{\infty} \gamma^l r_{t+l}$. The agent's policy, $\pi(a|s) : S \times A \rightarrow [0, 1]$, induces a state-action-value function (Q-function), $Q^\pi(s_t, a_t) = \mathbb{E}_{s_{t+1:\infty}, a_{t+1:\infty}} [R_t | s_t, a_t]$. Q-learning aims to estimate the optimal action-value function, $Q^*(x, a) = \max_\pi Q^\pi(x, a)$, via an estimated Q-function, $Q(s, a)$.

Training is carried out by collecting samples from the environment in order to obtain Monte-Carlo-Estimates of the expected return. For any sampled transition, the current estimate, $Q(s, a)$, is compared to a greedy one-step lookahead using the Bellman optimality operator, $\mathcal{T}Q(s, a) = \mathbb{E}_{s'} [r + \gamma \max_{a'} Q(s', a')]$. We can use samples to evaluate the Bellman update:

$$Q(s, a)_{k+1} = Q(s, a)_k + \alpha \left( r + \gamma \max_{a'} Q(s', a')_k - Q(s, a)_k \right)$$

Here $k$ is the iteration number, $\alpha$ is the learning rate and $r + \gamma \max_{a'} Q(s', a')_k - Q(s, a)_k$ is commonly referred to as the temporal-difference or TD-error. In the tabular case the Q-values for each state-action pair are maintained separately and the Bellman update is a contraction mapping. As a consequence at convergence this iterative process results in the optimal $Q$-function, $Q(s, a)_\infty = Q^*(s, a)$. Finally, $Q^*(s, a)$ trivially defines the optimal policy $\pi^*(x, a) = \delta(\arg\max_{a'} Q^*(x, a') - a)$, where $\delta(\cdot)$ is the Dirac-delta function. In contrast to the tabular case, DQN (Mnih et al., 2015) uses a neural network parametrized by a large number of weights, $\phi$, to represent the $Q$-function.

In order to reduce the variance of the update process the average square of the TD-error across a large number of transitions (the *batch*) is used as the DQN-loss. Using backpropagation the parameters of the neural network are updated to minimize the magnitude of the DQN error $\mathcal{L}(\phi) = \sum_{j=1}^{b} [(y_j^{DQN} - Q(x_j, a_j; \phi))^2]$. Here $y_j^{DQN} = u_j + \gamma \max_{a'_j} Q(x'_j, a'_j; \phi^-)$, is the target

function and $\phi^-$ is the target network, which contains a stale copy of the parameters. This target network helps to stabilize the training.

So far we have assumed that the agent has access to the Markov state, $s$, of the system. In a partially observable settings we augment the MDP with an observation space, $Z$, and observation function $O(s)$. In particular the observation $z \in Z$ is produced by the observation function $O(s) : S \to Z$. We further define an action-observation history $\tau \in T \equiv (Z \times U)^*$, which is used to condition a stochastic policy $\pi(a|\tau) : T \times U \to [0, 1]$. In recurrent deep RL (Hausknecht & Stone, 2015), this can be achieved using recurrent neural networks, such as LSTM (Hochreiter & Schmidhuber, 1997) or GRU (Cho et al., 2014) which we use here.

**Independent Q-Learning**   In multi-agent reinforcement learning, each agent $i \in N$ receives a private observation $O(x, i)$, where $i$ is the agent index and $O$ is the observation function. The agents also receive individual reward $r_t^i$ and take actions $a_t^i$. Furthermore the state-transition conditions on the joined action $\mathbf{a} \in \mathbf{A} \equiv A^n$.

In independent Q-learning each agent further estimates a Q-function $Q^i(\tau^i, a^i)$, treating the other agents and their policies as part of a non-stationary environment. A combination DQN with IQL commonly uses parameter-sharing across agents combined with an agents specific index in the observation function in order to accelerate learning while still allowing for specialization of policies.

## A.2   IMPLEMENTATION DETAILS

Our code has been submitted. Our neural network architecture implements the IQL architecture in Foerster et al. (2016).

