# OpenReview forum: "Modeling Fake News in Social Networks with Deep Multi-Agent Reinforcement Learning"
_ICLR.cc/2020/Conference — Reject_

### Official Review · AnonReviewer1 · 2019-10-21
**Official Blind Review #1**

**Rating:** 3

**Review:**

Update: I thank the authors for their response and for improving the paper. However, I maintain my position that the paper lacks a significant technical contribution to learning algorithms and that the applicability of the proposed approach is remains questionable in the current state.

Summary:
The paper proposes the  use of deep multi agent reinforcement learning (DMARL) for modelling fake news propagation and detection in social networks. The agents observe an informative yet noisy private signal and the actions of their neighbors (in the social network graph) and have to guess whether a claim (related to the received signal) is true or false. The fake news is modelled as an adversarial attack to the graph that either provides a hand-coded biased private signal to one of the agents or replaces one of  the agents with an RL policy trained to minimize the total reward of the agents in the graph (i.e. social network).

Main Comments:
I lean towards rejecting this paper because I do not find the methodological contribution to be significant enough to be published at ICLR, given that the main contribution is applying current techniques to a novel toy domain. While this paper attempts to apply DMARL to a new domain with real-world relevance, the authors only consider a toy example and make strong assumptions that are likely to break in the real world. Hence, it is not at all clear whether or how the conclusions of this paper would translate to more realistic scenarios of fake news in social networks. While strong assumptions and toy examples are reasonable for showing algorithmic improvements, this paper does not propose any improvement to core DMARL algorithms, but merely applies current methods to a new toy domain.

I am also concerned about the lack of comparison with other approaches to information aggregation in social networks. While I admit I am not familiar with that literature, I would still find it useful to provide some comparisons with non-DMARL (e.g. heuristic or game theoretic) approaches or at least some motivation for not comparing against those methods. The authors qualitatively describe those methods and their shortcomings, but the experimental section does not support those claims due to the lack of comparison. Despite all these concerns, the paper does indeed open-up numerous research directions and I can imagine follow-up papers being written that relax some of the current assumptions.

Other Questions / Comments:

1. Can you provide  some motivation for choosing to model the social network as a Barabsi Albert graph and why this is a reasonable modelling choice?

2. What happens if instead of a clustered or balanced graph, you have some combination of the two? It seems to me like that would be a more realistic scenario (i.e. a large graph containing subgraphs with different structures). Can the framework generalize to that? How would the conclusions change?

3. Is there any evidence that the conclusions supported by the experiments in this paper hold in real-world social networks and model some realistic aspects of social network dynamics?  Without such evidence, it is difficult to assess the relevance of this work for the  real-world application. Since the behavior of the agents in the graph is not guaranteed to be optimal / a best-response or even stable, is it at least a good approximation to human behavior in social networks? It would also be useful to show more comparisons against best-response or heuristic agents.

4. What is  the motivation behind considering a fixed budget of bias? Why not instead have a fixed number of agents that you will be biased? I think it would be informative to compare against applying beta = 3 to two citizens,  along with the focused and spread scenarios.
The legend in Figure 2 A & B is slightly confusing. I’d suggest using different styles for “private signal optimal” and “all signals optimal”.

5. Can you include error bars in Figure 2 D?

6. Can you provide results with a heuristic attacker that always lies about the claim? I read your intuition of why  you believe this wouldn’t be stronger than an attacker that is trained with RL together with the other agents, but is it actually true in practice, do the agents really learn to easily detect the “lying” attacker and distrust it? Based on your results, doesn’t it mean that one can find a heuristic attacker that has an equivalent behavior to the learned one ? Can one build even stronger hand-tuner attackers  based on heuristics?





**Experience Assessment:**

I have published one or two papers in this area.

**Review Assessment: Checking Correctness Of Derivations And Theory:**

I assessed the sensibility of the derivations and theory.

**Review Assessment: Checking Correctness Of Experiments:**

I assessed the sensibility of the experiments.

**Review Assessment: Thoroughness In Paper Reading:**

I read the paper at least twice and used my best judgement in assessing the paper.

---

> ### Author Response · Authors · 2019-11-09
> **Reply to Reviewer 1 (part 2)**
>
> 5. "What is the motivation behind considering a fixed budget of bias? Why not instead have a fixed number of agents that you will be biased? I think it would be informative to compare against applying beta = 3 to two citizens,  along with the focused and spread scenarios."
>
> In real world terms, a fixed bias budget corresponds to an attacker who has a fixed ad budget and can decide to either send a few ads to many agents or many ads to a few agents. In this sense the notion of a fixed biased budget is natural. In addition, we consider a fixed budget since, when agents are not trained in the presence of an attacker, the accuracy should be decreasing both in the bias and the number of agents attacked. When comparing the spread vs the focused attacks, we would like to isolate the effect of spreading an attack across agents (and thereby weakening it). If we were to set beta = 3 in the spread attack case we would trivally obtain a lower accuracy than in the focused attack (also with beta =3). Thus in order to move beyond this trivial result, we weaken the bias in the spread attack.
>
> The legend in Figure 2 A & B is slightly confusing. I’d suggest using different styles for “private signal optimal” and “all signals optima."
>
> We have adjusted the legend accordingly.
>
> 7. "Can you include error bars in Figure 2 D?"
>
> We include error bars in Figure 2 D based on the standard deviation of the accuracy across many runs of the game with different seeds but fixed neural network parameters. The error bars are so small they are not visible on the graph. However, we do not average across different runs of the training algorithm in this case.
>
> 8. "Can you provide results with a heuristic attacker that always lies about the claim? Based on your results, doesn’t it mean that one can find a heuristic attacker that has an equivalent behavior to the learned one ? Can one build even stronger hand-tuner attackers  based on heuristics?"
>
> Clearly, in the absence of training, an attacker that always lies would be optimal from the attacker's perspective. That is, if agents do not expect fake news, an attacker can optimally manipulate them by always lying. However, such a strategy would in fact be the worst possible attack strategy when agents are trained under attack.
>
> A heuristic attacker that always lies (and therefore must implicitly know the true state of the world), i.e. chooses action 1 if the theta = 0 and vice versa, would effectively provide a perfect signal to its neighbors. Agents will learn to exploit this and to choose the opposite action of this attacker. In this sense the attacker would not be an attacker at all, but instead would help agents discover the truth. Therefore, we believe a more useful benchmark is the random attacker; that is an agent who acts randomly and therefore reveals no information about the veracity of the claim.
>
> It may well be possible to replicate the attackers learned behavior with hand tuned heuristics. However, we believe that is precisely one of the strengths of our method that it does not require the development of hand tuned heuristics and that attack and information aggregation actions are learned via DMARL.
>
> References
> Barabási, Albert-László, and Réka Albert. "Emergence of scaling in random networks." science 286, no. 5439 (1999): 509-512.

---

> ### Author Response · Authors · 2019-11-09
> **Reply to Reviewer 1 (part 1)**
>
> We thank the reviewer for the valuable suggestions and comments. Below we provide answers to the points raised in the review.
>
> 1. "I am also concerned about the lack of comparison with other approaches to information aggregation in social networks. While I admit I am not familiar with that literature, I would still find it useful to provide some comparisons with non-DMARL (e.g. heuristic or game theoretic) approaches."
>
> We have added a comparison to the DeGroot heuristic to Figures 2 A-B and included a description of this heuristic. The comparison clearly indicates that our method outperforms this heuristic. We do not include a comparison to the game theoretic methods since these methods do not yield explicit solutions for the action update of the agents and can therefore not be simulated / computed. Instead, these methods only provide existence proofs of information aggregation in the limit.
>
> 2. "Can you provide some motivation for choosing to model the social network as a Barabsi Albert graph and why this is a reasonable modelling choice?"
>
> The Barabais-Albert graph has been commonly used to model many social networks, see Barabási (1999). As opposed to other random graph models, the Barabasi-Albert graph is able to generate clustering (i.e. triadic closure) which is an important feature of social networks.
>
> 3. "What happens if instead of a clustered or balanced graph, you have some combination of the two? It seems to me like that would be a more realistic scenario (i.e. a large graph containing subgraphs with different structures). Can the framework generalize to that? How would the conclusions change?"
>
> To some extent the clustered graph is "large" graph containing fully connected subgraphs and we do indeed consider the clustered graph a better description of social networks than the balanced graph. However, it is important to draw the comparison to the balanced graph in order to isolate the contribution of the community structure of the clustered graph. In addition, a social network might be able to intervene in the graph structure by strategically making link suggestions and thereby transform a clustered into a more balanced graph. Therefore comparison to the balanced graph is also important from a policy evaluation perspective. Having said that, we acknowledge that it would be useful to consider larger graphs with more complex community structures. We are currently exploring this direction for future work.
>
> 4. "Is there any evidence that the conclusions supported by the experiments in this paper hold in real-world social networks and model some realistic aspects of social network dynamics?  Without such evidence, it is difficult to assess the relevance of this work for the  real-world application. Since the behavior of the agents in the graph is not guaranteed to be optimal / a best-response or even stable, is it at least a good approximation to human behavior in social networks? It would also be useful to show more comparisons against best-response or heuristic agents."
>
> We fully agree that it is important to compare our results against human behavior. Therefore, we are currently planning experiments with human subjects in order to study how humans behave in the environment we studied with DMARL. This will allow us to evaluate to what extent the decision rules learned via DMARL are accurate.

---

### Official Review · AnonReviewer2 · 2019-10-25
**Official Blind Review #2**

**Rating:** 1

**Review:**

This paper proposes a model under which to study social networks under attacks attempting to propagate misinformation. It proposes a theoretical model based on assumptions on what kinds of graphs are common in social media and what kinds of attacks take place. While this could be interesting, the work presented falls short of what it promises, i.e. to develop a practical model of fake news on social networks,  because many of the assumptions made about the phenomena under study are unrealistic. In more detail

- There is a lot of research looking at social network graphs and analyzing them. As an example, here is a paper by Kate Starbird: https://www.aaai.org/ocs/index.php/ICWSM/ICWSM18/paper/view/17836/17028
I believe that there is little reason to generate data if one can collect them
- Even if generating the data can be justified, the graphs on which the methods are studies are 10-12 nodes. I can't see why such a low number was chosen given that the data is simulated, but it doesn't allow to assess whether the methods proposed would be practical in a real social network
- There is work suggesting that misinformation spreads faster than information: https://science.sciencemag.org/content/359/6380/1146
Thus it would make sense to take this into account in designing the graph theoretical model. Given this though, it is unlikely that the assumption that the social network will converge to the truth.
- Assuming that the social network graph remains fixed over time is also unrealistic. One can study how Twitter networks evolve over time
- The modelling of the users as merely voting on the truth or false value of a claim is not what happens in most social networks in which users fave/like, share, etc. Furthermore, it doesn't make that users want to prevent other from copying them. Being retweeted is a sign of influence, and users want to be influential.
- The attacks described do not seem to be grounded in any evidence/research on how misinformation propagates and what attackers actually do, so I can't accept the results of the analyses that use them.

**Experience Assessment:**

I have published in this field for several years.

**Review Assessment: Checking Correctness Of Derivations And Theory:**

N/A

**Review Assessment: Checking Correctness Of Experiments:**

I did not assess the experiments.

**Review Assessment: Thoroughness In Paper Reading:**

I made a quick assessment of this paper.

---

> ### Author Response · Authors · 2019-11-09
> **Reply to Reviewer 2 (part 2)**
>
> 5. "Assuming that the social network graph remains fixed over time is also unrealistic."
>
> Over the lifetime of a claim (e.g. a couple of hours to days), we believe that it is fair to assume the network is static as a benchmark. However, we are working on next steps that will allow for dynamic graphs.
>
> 6. "The modelling of the users as merely voting on the truth or false value of a claim is not what happens in most social networks."
>
> We acknowledge that users of social networks are motivated by many factors. The desire to be truthful is only one such factor. In order to keep the model simple and interpretable we focus on only one factor in this work. A natural and important extension would be to add a desire for conformity to the agents' incentive. Our model provides a benchmark for future research on this important aspect.
>
> 7. "Furthermore, it doesn't make sense that users want to prevent others from copying them. Being retweeted is a sign of influence, and users want to be influential."
>
> Again, we simplify agents' incentives for the sake of interpretability. However, note that retweeting is not necessarily equivalent to copying in our model. Copying amounts to imitating the actions of others, which does not necessarily confer status to the agent whose actions are copied.
>
> 8. "The attacks described do not seem to be grounded in any evidence/research on how misinformation propagates and what attackers actually do, so I can't accept the results of the analyses that use them."
>
> We believe that our model of attackers captures a number of important features of real world fake news attacks. For example, in real world fake news attacks, perpetrators sent targeted ads (see for example Chiou et al. (2018)) to members of social networks to provide them with biased information about a claim. This corresponds one-to-one to agents receiving biased signals in our model. We have added this reference to the paper and thank the reviewer for pointing out this shortcoming.
>
> References
> Chiou, Lesley, and Catherine Tucker. Fake news and advertising on social media: A study of the anti-vaccination movement. No. w25223. National Bureau of Economic Research, 2018.

---

> ### Author Response · Authors · 2019-11-09
> **Reply to Reviewer 2 (part 1)**
>
> We would like to thank the reviewer for the valuable comments and suggestions. Below we respond to the points raised in the review.
>
> 1. "The work presented falls short of what it promises, i.e. to develop a practical model of fake news on social networks, because many of the assumptions made about the phenomena under study are unrealistic."
>
> As outlined in our general comments above, we did not intend to claim that our paper provides a practical method for the detection or prevention of fake news on social networks. When using the term “practical” we were instead referring to a model that is easy to use, modify and run. We understand that this has created some confusion and have updated the wording of the paper accordingly.
>
> Indeed, the development of a model for the detection or prevention of fake news would be a very ambitious task and would go far beyond the current state of the literature. All models of fake news on social networks need to make abstractions of some kind and we acknowledge that some of the assumptions made in our model may appear particularly strong. However, as explained above, we choose to study DMARL for fake news within a particularly simple model as a first step to demonstrate the usefulness of the method. In addition, while our model may be simple, the agents’ learned decision rules are not and in fact are likely to be more realistic than ad-hoc heuristics.
>
> 2. "I believe that there is little reason to generate data if one can collect them."
>
> Respectfully, we disagree. It is very difficult to make counterfactual predictions based on observational data of social phenomena. Asking the question: "what happens when agents learn to adapt to the presence of fake news?" with observational data is very difficult if not impossible. The only sound data driven approach would be costly large scale randomized control trials. However, we need to be able to evaluate precisely questions of this type to design effective policies against fake news. This is why we need models. Again, we acknowledge that our model is simple and that its conclusions need to be further validated, but observational studies alone, even if the data were available to scientists, are unlikely to provide the full picture.
>
> 3. "I can't see why such a low number was chosen given that the data is simulated, but it doesn't allow to assess whether the methods proposed would be practical in a real social network."
>
> We acknowledge that our networks are small and, as we note in our conclusion, work towards improving our method to scale to larger networks. Note that DMARL computational complexity (with recurrent neural nets) is roughly proportional to the number of time steps and agents in the game. The number of time steps again should scale roughly proportionally to the number of agents. Therefore, our current method is quadratic in the number of agents. Training agents to learn policies with DMARL takes time and given our currently limited computational resources we have chosen to showcase our method for small networks. In principle, with sufficient computing capacity our experiments can be generalized to a large numbers of agents. We plan to do this in future work. Also in future work, we plan to improve our method to reduce time complexity and improve scaling.
>
> 4. "Faster spreading of fake news: Given this though, it is unlikely that the assumption that the social network will converge to the truth."
>
> First, note that we do not assume that the social network converges to the truth. Convergence is an "emergent" property of the actions of the agents which learn both from private signals and their neighbors actions. Also note that our model does not speak to the speed of spreading of a news story on the network. All agents are instantly aware of a claim made and respond to the claim by stating whether or not they believe the claim is true or false. In addition, the spread of a piece of news (i.e. it's virality) tells us relatively little about people's beliefs about its veracity. A piece of fake news can be shared simply because of comedy value or because people believe it is outrageous. In our model we are interested in the effectiveness of a piece of fake news not measured in how many people it reaches but how many people believe it is true. Having said this, modeling the spread of fake news on social networks is important and we hope to address this point in future work.

---

### Official Review · AnonReviewer3 · 2019-11-04
**Official Blind Review #3**

**Rating:** 1

**Review:**

In this work, the authors aim to solve the problem of fake news detection in social media. The proposed method is built upon a multi-agent reinforcement learning method. Although the problem of fake news detection in social media has been extensively studied and many method including deep learning have been investigated to help solve the problem, it is relatively novel to use multi-agent reinforcement learning in this field. The proposed method is based on traditional multi-agent deep reinforcement learning approach and the authors extend the conventional framework by introducing the role of attacking agents - agents that can spread biased information or even take over the stance of regular users. The paper has been well written and necessary details for reproducing the experimental results have been provided with a link to the code repository. However, a major concern of mine is the contribution of novelty of the manuscript.

The main contribution of novelty of the work, as claimed by the authors, is that they come up with a practical solution for fake news detection with deep reinforcement learning. Most research efforts have been focusing on detecting fake and deep-fake content while very few pay attention to utilizing machine learning in learning a best action/practice. This is because the root cause of the widespread of fake news is quite complicated. Besides those who intentionally create and spread fake news and biased content, the innocent users' major problem is how to quickly identify fake news from massive amount of information flows. The idea that building a practical fake news prevention solution by using multi-agent reinforcement learning seems to have underestimated the complexity of the misinformation challenge. For example, three high-level suggestions/solutions proposed in the manuscript include social network users should be more aware of the presence of fake news, keeping private information private on social networks, and encouraging well balanced social network structures. My question is that how we can apply these solutions in the real world? Therefore, I think this piece of work is more theoretical rather than practical.

In terms of the technical part, the authors propose to introduce agents for fake news and biased information. The technical solution is solely based on multi-agent reinforcement learning and the extension is straightforward. The assumption that there is only one kind of role for fake news dissemination in social networks again underestimates the complexity of the problem in the real world. E.g., there are users who intentionally create fake news in social networks, and also users who are not aware of a piece of information being fake and yet still deeply believe what they spread is true. I would suggest the authors to find more related work in the field of information diffusion, where researchers have long been focusing on competitive information propagation in social networks with multiple parties (such as political campaign and word-of-mouth social marketing).

In conclusion, I think the work is well-written and quite interesting - solving an emerging and important problem from a new perspective. However, both the hypothesis and the technical solution are lacking enough contributions of novelty. I would like to suggest the authors either make the solution actually practical or focus more on the theoretical part of competitive information diffusion in social networks.

**Experience Assessment:**

I have published in this field for several years.

**Review Assessment: Checking Correctness Of Derivations And Theory:**

I assessed the sensibility of the derivations and theory.

**Review Assessment: Checking Correctness Of Experiments:**

I carefully checked the experiments.

**Review Assessment: Thoroughness In Paper Reading:**

I read the paper thoroughly.

---

> ### Author Response · Authors · 2019-11-09
> **Reply to Reviewer 3**
>
> We thank the reviewer for the valuable comments and feedback. Below, we address the concerns raised in the review.
>
> 1. "My question is that how we can apply these solutions in the real world? Therefore, I think this piece of work is more theoretical rather than practical."
>
> We agree that our work is theoretical and provides qualitative conclusions which form the starting point for future research. When using the term “practical” we were instead referring to a model that is easy to use, modify and run. We understand that this has created some confusion and have updated the wording of the paper accordingly. Before decision makers should act on our conclusions, they should be validated in more realistic models. Nevertheless such qualitative conclusions are a useful and necessary first step. We believe that research should progress from simple to complex models. A model with too many parameters and moving parts can be difficult to interpret and understand. Therefore it is important to first build simple models, such as ours, to provide relevant benchmarks and further intuition for more complex models.
>
> 2. "The assumption that there is only one kind of role for fake news dissemination in social networks again underestimates the complexity of the problem in the real world. E.g., there are users who intentionally create fake news in social networks, and also users who are not aware of a piece of information being fake and yet still deeply believe what they spread is true. I would suggest the authors to find more related work in the field of information diffusion, where researchers have long been focusing on competitive information propagation in social networks with multiple parties (such as political campaigns and word-of-mouth social marketing). "
>
> We agree that our model abstracts from many important aspects of fake news dissemination. However, our model does capture the cases mentioned above: the users who deliberately create fake news and users who believe a false claim is true. The attacker in our model can be seen as the creator of fake news. Agents who believe a false claim is true have, by chance, received a private signal that supports the veracity of a false claim. The frequency with which this happens depends on the variance of the private signal. In our choice of parameters this occurs with a roughly 31% probability.
>
> We would like to highlight the difference in focus between our work and the literature on competitive information diffusion. We focus on the decision making of the users of the social network, i.e. we model the diffusion process. We also begin to model the attacker in this work and in future work plan to extend our efforts in this direction. The literature on competitive information diffusion takes the diffusion process as given and focuses on the decisions of the attacker (those who seed the diffusion process). This is an important and interesting question which is naturally complementary to our work. We thank the reviewer for pointing out this connection and, as mentioned above, plan to extend our model to improve the way the attackers are modeled.

---

> > ### Comment · AnonReviewer3 · 2019-11-09
> > **response to authors**
> >
> > Thanks for the feedback and I appreciate the efforts of improving the work. However, there is an obvious consensus among the reviewers that the manuscript is far from being applicable in the real world. I would suggest the authors find more relevant venues for future submissions, such as social science and network science conferences that emphasize more on the application of fake news detection and theoretical analyses.

---

### Author Response · Authors · 2019-11-09
**General comments to reviewers: existing literature**

Existing literature:
Heuristic models of belief propagation (such as DeGroot (1974)) or models of competitive information diffusion (Bulteau (2015)) do not include the forces (truth seeking and adaptation) mentioned above and tend to focus on different aspects of the information aggregation / diffusion process.

For example, in models of competitive information diffusion, agents (i.e. the users of the social network) are essentially subsumed into a threshold diffusion process. The focus of these models is on the optimal strategy of those who seed information into the network which loosely correspond to the attackers in our model. We instead focus on how agents choose to respond to claims; that is, we model the diffusion process rather than assuming a process ex-ante. Our approach and the literature on competitive information diffusion are therefore natural complements.

In this paper, we begin to incorporate the work on competitive information diffusion by allowing the attacker to learn an attack strategy. A natural next step, which we plan to address in future work, is to allow the attacker to also choose which subset of agents to attack. Of course, this can become difficult once the attacker can choose more than one agent to attack, due to the combinatorial explosion of the action space.

References
DeGroot, Morris H. "Reaching a consensus." Journal of the American Statistical Association 69, no. 345 (1974): 118-121.

Bulteau, L., Froese, V. and Talmon, N., 2015, May. Multi-Player diffusion games on graph classes. In International Conference on Theory and Applications of Models of Computation (pp. 200-211). Springer, Cham.

---

### Author Response · Authors · 2019-11-09
**General comments to reviewers: contribution**

We would like to thank the reviewers for the comments and valuable feedback on this submission. Before responding to each reviewer individually, we would like to clarify the contribution of our paper and its relation to the existing literature in order to address some common concerns raised by the reviewers.

Contribution:
Our contribution is twofold.

1. A novel application of DMARL: We demonstrate how deep multi-agent reinforcement learning (DMARL) can be helpful in the modeling of fake news on social networks. In particular, DMARL allows us to study the emergent behavior of agents in models which are analytically intractable without resorting to hand-crafted heuristics. This is important since (i) only the simplest models are analytically tractable and (ii) heuristic methods have to be redesigned for every model variation, whereas DMARL provides a generic solution method.

Our main aim here is to highlight the usefulness of DMARL in the domain of fake news modelling. We do not intend or claim that our paper provides a practical, data driven method for the detection or prevention of fake news on social networks, but rather develop a necessary precursor for such a massive undertaking . When using the term “practical” we instead refer to a model that is easy to use, modify and run. That is to be contrasted with analytical methods for solving models of utility maximizing agents which are very difficult to apply and yield results only for very specific and stylized scenarios. We understand that this has caused some confusion and have updated the wording of the paper accordingly.

We seek to demonstrate the usefulness of DMARL in the simplest possible model of fake news (with utility maximizing agents) which has clear analytical benchmarks. We believe that this is a natural and necessary first step before more complex and realistic models can be addressed. We have further updated the paper to show that our method outperforms existing hand-coded heuristics for information aggregation.

2. Qualitative insights derived from experiments: We acknowledge that our model is simple and abstracts from many important real world features. Nevertheless, our model captures important forces in the propagation of fake news in reduced form which allow us to derive qualitative statements. These forces are:

(i) agents in the general population want to act in accordance with the truth. Naturally, there are also actors which are not interested in the truth. However, we believe that in general people do not want to be wrong, either because of the social stigma associated with supporting false claims or because of an intrinsic motivation. It is important to note here that our model focuses on the belief about claims which can be factually verified. It is not a model about voting behavior (such as the classic voter models) or opinion formation (such as moral convictions). We believe that a realistic model of fake news needs to be linked back to the ground truth via the agents' incentives.

(ii) agents try to form beliefs by combining their observations in a context-dependent manner. This implies that agents do not follow static decision rules but learn to adapt to changing circumstances. As fake news is injected into the system, agents learn to "detect" it and adjust their beliefs accordingly.

Incorporating these forces allows us to draw qualitative conclusions on how agents can adapt to fake news attacks and how the network structure can affect information aggregation and the effectiveness of attack. We acknowledge that these are qualitative conclusions which eventually need to be corroborated by more realistic models. Nevertheless, we have reason to expect that these conclusions generalize to larger networks and more complex attack models. We plan to confirm this conjecture in future work.

---

### Author Response · Authors · 2019-11-09
**Summary of changes in revised version**

We have made the following edits to the paper:

1. We have clarified our contribution and motivation.
2. We have updated Figure 2 to incorporate a direct comparison to heuristic benchmarks.
3. We have fixed the legend of Figure 2 (A-B).
4. We have added a discussion of the literature on competitive information diffusion.

---

### Decision · Program_Chairs · 2019-12-19

**Decision:**

Reject

**Comment:**

The paper aims to model fake news by drawing tools from multi-agent reinforcement learning. After the discussion period, there is a consensus among the reviewers that the paper lacks novel technical contributions. The reviewers also acknowledge that paper also doesn't quite deliver a practical solution as claimed by the authors.